# Ridge Regression and Provable Deterministic Ridge Leverage Score Sampling

**Shannon R. McCurdy**
California Institute for Quantitative Biosciences
UC Berkeley
Berkeley, CA 94702
smccurdy@berkeley.edu

## Abstract

Ridge leverage scores provide a balance between low-rank approximation and regularization, and are ubiquitous in randomized linear algebra and machine learning. Deterministic algorithms are also of interest in the moderately big data regime, because deterministic algorithms provide interpretability to the practitioner by having no failure probability and always returning the same results. We provide provable guarantees for deterministic column sampling using ridge leverage scores. The matrix sketch returned by our algorithm is a column subset of the original matrix, yielding additional interpretability. Like the randomized counterparts, the deterministic algorithm provides $(1+\epsilon)$ error column subset selection, $(1+\epsilon)$ error projection-cost preservation, and an additive-multiplicative spectral bound. We also show that under the assumption of power-law decay of ridge leverage scores, this deterministic algorithm is provably as accurate as randomized algorithms. Lastly, ridge regression is frequently used to regularize ill-posed linear least-squares problems. While ridge regression provides shrinkage for the regression coefficients, many of the coefficients remain small but non-zero. Performing ridge regression with the matrix sketch returned by our algorithm and a particular regularization parameter forces coefficients to zero and has a provable $(1+\epsilon)$ bound on the statistical risk. As such, it is an interesting alternative to elastic net regularization.

## 1 Introduction

Classical leverage scores quantify the importance of each column $i$ for the range space of the sample-by-feature data matrix $\mathbf{A} \in \mathbb{R}^{n \times d}$. Classical leverage scores have been used in regression diagnostics, outlier detection, and randomized matrix algorithms (Velleman and Welsch, 1981; Chatterjee and Hadi, 1986; Drineas et al., 2008). Historically, leverage scores were used to select informative samples (rows, in our matrix orientation). More recently, as datasets with $d > n$ have become more common, leverage scores have been used to select informative features (columns, in our matrix orientation). There are many different flavors of leverage scores, and we will focus on a variation called ridge leverage scores. However, to appreciate the advantages of ridge leverage scores, we also briefly review classical and rank-$k$ subspace leverage scores.

Ridge leverage scores were introduced by Alaoui and Mahoney (2015) to give statistical bounds for the Nyström approximation for kernel ridge regression. Alaoui and Mahoney (2015) argue that ridge leverage scores provide the relevant notion of leverage in the context of kernel ridge regression. Ridge leverage scores have been successfully used in kernel ridge regression to approximate the symmetric kernel matrix ($\in \mathbb{R}^{n \times n}$) by selecting informative samples (Alaoui and Mahoney, 2015; Rudi et al., 2015). Cohen et al. (2017) provide a definition for ridge leverage scores for selecting informative features from the non-symmetric sample-by-feature data matrix $\mathbf{A} \in \mathbb{R}^{n \times d}$. The ridge

leverage score $\bar{\tau}_i(\mathbf{A})$ for the $i^{th}$ column of $\mathbf{A}$ is,

$$\bar{\tau}_i(\mathbf{A}) = \mathbf{a}_i^T \left(\mathbf{A}\mathbf{A}^T + \lambda_2 \mathbf{I}\right)^+ \mathbf{a}_i, \tag{1}$$

where the $i^{th}$ column of $\mathbf{A}$ is an $(n \times 1)$-vector denoted by $\mathbf{a}_i$, $\mathbf{M}^+$ denotes the Moore-Penrose pseudoinverse of $\mathbf{M}$, and $\lambda_2$ is the regularization parameter. We will always choose $\lambda_2 = \frac{1}{k}||\mathbf{A} - \mathbf{A}_k||_F^2$, where $\mathbf{A}_k$ is the rank-$k$ SVD approximation to $\mathbf{A}$, defined in Sec. 1.2, because this choice of regularization parameter gives the stated guarantees. In contrast to ridge leverage scores, the rank-$k$ subspace leverage score $\tau_i(\mathbf{A}_k)$ is,

$$\tau_i(\mathbf{A}_k) = \mathbf{a}_i^T (\mathbf{A}_k \mathbf{A}_k^T)^+ \mathbf{a}_i. \tag{2}$$

The classical leverage score is the ridge leverage score (and also the rank-$k$ subspace leverage score) evaluated at $k = \text{rank}(\mathbf{A}) = r \leq n$.

Ridge leverage scores and rank-$k$ subspace leverage scores take two different approaches to mitigating the small singular values components of $\mathbf{A}\mathbf{A}^T$ in classical leverage scores. Ridge leverage scores diminish the importance of small principle components through regularization, as opposed to rank-$k$ subspace leverage scores, which omit the small principle components entirely. Cohen et al. (2017) argue that regularization is a more natural and stable alternative to omission. For randomized algorithms with ridge leverage score sampling, Cohen et al. (2017) prove bounds for the spectrum, column subset selection, and projection-cost preservation (counterparts to our Theorems 1, 2, and 3 for deterministic ridge leverage scores, respectively). The first and the last bounds hold for a weighted column subset of the full data matrix. These bounds require $O(k \log(k/\delta)/\epsilon^2)$ columns, where $\delta$ is the failure probability and $\epsilon$ is the error.

In the "big data" era, much attention has been paid to randomized algorithms due to improved algorithm performance and ease of generalization to the streaming setting. However, for moderately big data (i.e. the feature set is too large for inspection by humans, but the algorithm performance is not a limitation), deterministic algorithms provide more interpretability to the practitioner than randomized algorithms, since they always provide the same results and have no failure probability.

The usefulness of deterministic algorithms has already been recognized. Papailiopoulos et al. (2014) introduce a deterministic algorithm for sampling columns from rank-$k$ subspace leverage scores and provide a columns subset selection bound (the counterpart to our Theorem 2 for deterministic ridge leverage scores). McCurdy et al. (2017) prove a $(1 + \epsilon)$ spectral bound for Papailiopoulos et al. (2014)'s deterministic algorithm and for random sampling with rank-$k$ subspace leverage scores (the counterpart to our Theorem 1 for deterministic ridge leverage scores). One major drawback of using the rank-$k$ subspace leverage scores is that their relative spectral bound is limited to the rank-$k$ subspace projection of the column subset matrix $\mathbf{C}$ and the full data matrix $\mathbf{A}$, so to get a relative spectral bound on the complete subspace requires $k = n$. A consequence of this is that projection-cost preservation also requires $k = n$ (the counterpart to our Theorem 3). One advantage of using deterministic rather than randomized rank-$k$ subspace leverage score algorithms is that under the condition of power-law decay in the sorted rank-$k$ subspace leverage scores, the deterministic algorithm chooses fewer columns than random sampling with the same error for the column subset selection bound when $\max((2k/\epsilon)^{\frac{1}{a}} - 1, (2k/((a-1)\epsilon))^{\frac{1}{a-1}} - 1, k) < Ck \log(k/\delta)/\epsilon^2$, where $a$ is the decay power and $C$ is an absolute constant (Papailiopoulos et al., 2014) (this is the counterpart to our Theorem 5). In addition, Papailiopoulos et al. (2014) show that many real datasets display power-law decay in the sorted rank-$k$ subspace leverage scores, illustrating the deterministic algorithm's real-world utility.

Ridge regression (Hoerl and Kennard, 1970) is a commonly used method to regularize ill-posed linear least-squares problems. The ridge regression minimization problem is, for outcome $\mathbf{y} \in \mathbb{R}^n$, features $\mathbf{A} \in \mathbb{R}^{n \times d}$, and coefficients $\mathbf{x} \in \mathbb{R}^d$,

$$\begin{aligned}
\hat{\mathbf{x}}_\mathbf{A} &= \underset{\mathbf{x}}{\text{argmin}} \left(||\mathbf{y} - \mathbf{A}\mathbf{x}||_2^2 + \lambda_2 ||\mathbf{x}||_2^2\right) \\
&= \left(\mathbf{A}^T \mathbf{A} + \lambda_2 \mathbf{I}\right)^{-1} \mathbf{A}^T \mathbf{y}.
\end{aligned} \tag{3}$$

where the regularization parameter $\lambda_2$ penalizes the size of the coefficients in the minimization problem. We will always choose $\lambda_2 = \frac{1}{k}||\mathbf{A} - \mathbf{A}_k||_F^2$ for ridge regression with matrix $\mathbf{A}$.

In ridge regression, the underlying statistical model for data generation is,

$$\mathbf{y} = \mathbf{y}^* + \sigma^2 \boldsymbol{\xi}, \tag{4}$$

where $\mathbf{y}^* = \mathbf{A}\mathbf{x}^*$ is a deterministic linear function of the fixed design features $\mathbf{A}$ and $\boldsymbol{\xi} \sim \mathcal{N}(0, \mathbf{I})$ is the random error. The mean squared error is a measure of statistical risk $\mathcal{R}(\hat{\mathbf{y}})$ for the squared error loss function and estimator $\hat{\mathbf{y}}$ and is,

$$\mathcal{R}(\hat{\mathbf{y}}) = \tfrac{1}{n}\mathbb{E}_{\boldsymbol{\xi}}\left[||\hat{\mathbf{y}} - \mathbf{y}^*||_2^2\right]. \tag{5}$$

Ridge regression is often chosen over regression subset selection procedures for regularization because, as a continuous shrinkage method, it exhibits lower variability (Breiman, 1996). However many ridge regression coefficients can be small but non-zero, leading to a lack of interpretability for moderately big data ($d > n$). The lasso method (Tibshirani, 1994) provides continuous shrinkage and automatic feature selection using an $L_1$ penalty function instead of the $L_2$ penalty function in ridge regression, but for $d > n$ case, lasso saturates at $n$ features. The elastic net algorithm combines lasso ($L_1$ penalty function) and ridge regression ($L_2$ penalty function) for continuous shrinkage and automatic feature selection (Zou and Hastie, 2005).

## 1.1 Contributions

We explore deterministic ridge leverage score (DRLS) sampling for matrix approximation and for feature selection in concert with ridge regression. This work has two main motivations: (1) the advantages of ridge leverage scores over rank-$k$ subspace leverage scores, and (2) the advantages of deterministic algorithms in some practical settings. This work complements Papailiopoulos et al. (2014), who considered deterministic rank-$k$ subspace leverage sampling and experiments on real data, but did not consider DRLS sampling or uses beyond matrix approximation. This work also complements Cohen et al. (2017), who considered randomized RLS sampling but did not consider DRLS sampling, the uses of RLS sampling beyond matrix approximation (e.g. ridge regression), or experiments on real data.

We introduce a deterministic algorithm (Algorithm 1) for ridge leverage score sampling inspired by the deterministic algorithm for rank-$k$ subspace leverage score sampling (Papailiopoulos et al., 2014). By using ridge leverage scores instead of rank-$k$ subspace scores in the deterministic algorithm, we prove significantly better bounds for the column subset matrix $\mathbf{C}$ (see Table 1 for a comparison). We prove that the same additive-multiplicative spectral bounds (Theorem 1), $(1 + \epsilon)$ columns subset selection (Theorem 2), and $(1+\epsilon)$ projection-cost preservation (Theorem 3) hold for DRLS column sampling as for random sampling as in Cohen et al. (2017). We show that under the condition of power-law decay in the ridge leverage scores, the deterministic algorithm chooses fewer columns than random sampling with the same error when $\max((4k/\epsilon)^{\frac{1}{a}} - 1, (4k/((a-1)\epsilon))^{\frac{1}{a-1}} - 1, k) < Ck\log(k/\delta)/\epsilon^2$, where $a$ is the decay power and $C$ is an absolute constant (Theorem 5).

We combine deterministic ridge leverage score column subset selection with ridge regression for a particular value of the regularization parameter, providing automatic feature selection and continuous shrinkage. This procedure has a provable $(1 + \epsilon)$ bound on the statistical risk (Theorem 4). The proof techniques are such that a $(1 + \epsilon)$ bound on the statistical risk also holds for randomized ridge leverage score sampling. Our ridge regression theorem is novel to both deterministic and randomized sampling with ridge leverage scores (as far as we know, this has never been considered for any leverage score), another demonstrable advance of the state of the art, and one of our main results.

We also provide a proof-of-concept illustration on real biological data, with figures included in the Supplementary Materials. Our real-data illustration makes a strong case for the empirical usefulness of the DRLS algorithm and bounds. The real data exhibits striking power law decay of the ridge leverage scores (Figure 7), justifying the assumptions underlying the use of DRLS sampling (Theorem 5).

Our work is triply beneficial from the interpretability standpoint; it is deterministic, it chooses a subset of representative columns, and it comes with four desirable error guarantees for all rank-$k$, three of which stem from naturalness of the low-rank ridge regularization.

## 1.2 Notation

The *singular value decomposition* (SVD) of any complex matrix $\mathbf{A}$ is $\mathbf{A} = \mathbf{U\Sigma V}^\dagger$, where $\mathbf{U}$ and $\mathbf{V}$ are square unitary matrices ($\mathbf{U}^\dagger\mathbf{U} = \mathbf{U}\mathbf{U}^\dagger = \mathbf{I}$, $\mathbf{V}^\dagger\mathbf{V} = \mathbf{V}\mathbf{V}^\dagger = \mathbf{I}$), $\mathbf{\Sigma}$ is a rectangular diagonal matrix with real non-negative non-increasingly ordered entries. $\mathbf{U}^\dagger$ is the complex conjugate and

Table 1: Comparison of deterministic ridge and rank-$k$ subspace leverage score theorems.

| Deterministic Sampling Algorithm | Rank-$k$ Subspace Papailiopoulos et al. (2014) | Rank-$k$ Ridge Algorithm 1 |
|---|---|---|
| Spectral Bound for $\mathbf{CC}^T$ | Multiplicative, $k = n$ McCurdy et al. (2017) | Additive-Multiplicative, all $k$ Theorem 1 |
| Column Subset Selection | all $k$ Papailiopoulos et al. (2014) | all $k$ Theorem 2 |
| Rank-$k$ Projection Cost Preservation | $k = n$ | all $k$ Theorem 3 |
| Approximate Ridge Regression Risk | N/A | all $k$ Theorem 4 |
| Leverage Power-law Decay | all $k$ Papailiopoulos et al. (2014) | all $k$ Theorem 5 |

transpose of $\mathbf{U}$, and $\mathbf{I}$ is the identity matrix. The diagonal elements of $\mathbf{\Sigma}$ are called the *singular values*, and they are the positive square roots of the eigenvalues of both $\mathbf{AA}^\dagger$ and $\mathbf{A}^\dagger \mathbf{A}$, which have eigenvectors $\mathbf{U}$ and $\mathbf{V}$, respectively. $\mathbf{U}$ and $\mathbf{V}$ are the *left* and *right singular vectors* of $\mathbf{A}$.

Defining $\mathbf{U}_k$ as the first $k$ columns of $\mathbf{U}$ and analogously for $\mathbf{V}$, and $\mathbf{\Sigma}_k$ the square diagonal matrix with the first $k$ entries of $\mathbf{\Sigma}$, then $\mathbf{A}_k = \mathbf{U}_k \mathbf{\Sigma}_k \mathbf{V}_k^\dagger$ is the rank-$k$ SVD approximation to $\mathbf{A}$. Furthermore, we refer to matrix with only the last $n - k$ columns of $\mathbf{U}, \mathbf{V}$ and last $n - k$ entries in $\mathbf{\Sigma}$ as $\mathbf{U}_{\backslash k}, \mathbf{V}_{\backslash k}$, and $\mathbf{\Sigma}_{\backslash k}$.

The Moore-Penrose pseudo inverse of a rank $k$ matrix $\mathbf{A}$ is given by $\mathbf{A}^+ = \mathbf{V}_k \mathbf{\Sigma}_k^{-1} \mathbf{U}_k^\dagger$.

The Frobenius norm $||\mathbf{A}||_F$ of a matrix $\mathbf{A}$ is given by $||\mathbf{A}||_F^2 = \mathrm{tr}\left(\mathbf{AA}^\dagger\right)$. The spectral norm $||\mathbf{A}||_2$ of a matrix $\mathbf{A}$ is given by the largest singular value of $\mathbf{A}$.

## 2 Deterministic Ridge Leverage Score (DRLS) Column Sampling

### 2.1 The DRLS Algorithm

**Algorithm 1.** *The DRLS algorithm selects for the submatrix $\mathbf{C}$ all columns $i$ with ridge leverage score $\bar{\tau}_i(\mathbf{A})$ above a threshold $\theta$, determined by the error tolerance $\epsilon$. This algorithm is deeply indebted to the deterministic algorithm of Papailiopoulos et al. (2014). It substitutes ridge leverage scores for rank-$k$ subspace scores, and has a different stopping parameter. The algorithm is as follows.*

1. *Choose the error tolerance, $\epsilon$.*

2. *For every column $i$, calculate the ridge leverage scores $\bar{\tau}_i(\mathbf{A})$ (Eqn. 1).*

3. *Sort the columns by $\bar{\tau}_i(\mathbf{A})$, from largest to smallest. The sorted column indices are $\pi_i$.*

4. *Define an empty set $\Theta = \{\}$. Starting with the largest sorted column index $\pi_0$, add the corresponding column index $i$ to the set $\Theta$, in decreasing order, until,*

$$\sum_{i \in \Theta} \bar{\tau}_i(\mathbf{A}) > \bar{t} - \epsilon, \tag{6}$$

*and then stop. Note that $\bar{t} = \sum_{i=1}^d \bar{\tau}_i(\mathbf{A}) \le 2k$ (see Sec.1.2 for proof). It will be useful to define $\tilde{\epsilon} = \sum_{i \notin \Theta} \bar{\tau}_i(\mathbf{A})$. Eqn. 6 can equivalently be written as $\epsilon > \tilde{\epsilon}$.*

5. *If the set size $|\Theta| < k$, continue adding columns in decreasing order until $|\Theta| = k$.*

6. *The leverage score $\bar{\tau}_i(\mathbf{A})$ of the last column $i$ included in $\Theta$ defines the leverage score threshold $\theta$.*

7. *Introduce a rectangular selection matrix $\mathbf{S}$ of size $d \times |\Theta|$. If the column indexed by $(i, \pi_i)$ is in $\Theta$, then $\mathbf{S}_{i,\pi_i} = 1$. $\mathbf{S}_{i,\pi_i} = 0$ otherwise. The DRLS submatrix is $\mathbf{C} = \mathbf{AS}$.*

*Note that when the ridge leverage scores on either side of the threshold are not equal, the algorithm returns a unique solution. Otherwise, there are as many solutions as there are columns with equal ridge leverage scores at the threshold.*

Algorithm 1 requires $O(\min(d, n)nd)$ arithmetic operations.

## 3  Approximation Guarantees

### 3.1  Bounds for DRLS

We derive a new additive-multiplicative spectral approximation bound (Eqn. 7) for the square of the submatrix $\mathbf{C}$ selected with DRLS.

**Theorem 1.** *Additive-Multiplicative Spectral Bound: Let $\mathbf{A} \in \mathbb{R}^{n \times d}$ be a matrix of at least rank $k$ and $\bar{\tau}_i(\mathbf{A})$ be defined as in Eqn. 1. Construct $\mathbf{C}$ following the DRLS algorithm described in Sec. 2.1. Then $\mathbf{C}$ satisfies,*

$$(1 - \epsilon)\mathbf{A}\mathbf{A}^T - \frac{\epsilon}{k}||\mathbf{A}_{\backslash k}||_F^2 \mathbf{I} \quad \preceq \quad \mathbf{C}\mathbf{C}^T \preceq \mathbf{A}\mathbf{A}^T. \tag{7}$$

*The symbol $\preceq$ denotes the Loewner partial ordering which is reviewed in Sec 1.1 (see Horn and Johnson (2013) for a thorough discussion).*

Conceptually, the Loewner ordering in Eqn. 7 is the generalization of the ordering of real numbers (e.g. $1 < 1.5$) to Hermitian matrices. Statements of Loewner ordering are quite powerful; important consequences include inequalities for the eigenvalues. We will use Eqn. 7 to prove Theorems 2, 3, and 4. Note that our additive-multiplicative bound holds for an un-weighted column subset of $\mathbf{A}$.

**Theorem 2.** *Column Subset Selection: Let $\mathbf{A} \in \mathbb{R}^{n \times d}$ be a matrix of at least rank $k$ and $\bar{\tau}_i(\mathbf{A})$ be defined as in Eqn. 1. Construct $\mathbf{C}$ following the DRLS algorithm described in Sec. 2.1. Then $\mathbf{C}$ satisfies,*

$$||\mathbf{A} - \mathbf{C}\mathbf{C}^+\mathbf{A}||_F^2 \leq ||\mathbf{A} - \mathbf{\Pi}_{\mathbf{C},k}^F(\mathbf{A})||_F^2 \leq (1 + 4\epsilon)||\mathbf{A}_{\backslash k}||_F^2, \tag{8}$$

*with $\quad 0 < \epsilon < \frac{1}{4}$ and where $\mathbf{\Pi}_{\mathbf{C},k}^F(\mathbf{A}) = (\mathbf{C}\mathbf{C}^+\mathbf{A})_k$ is the best rank-$k$ approximation to $\mathbf{A}$ in the column space of $\mathbf{C}$ with the respect to the Frobenius norm.*

Column subset selection algorithms are widely used for feature selection for high-dimensional data, since the aim of the column subset selection problem is to find a small number of columns of $\mathbf{A}$ that approximate the column space nearly as well as the top $k$ singular vectors.

**Theorem 3.** *Rank-$k$ Projection-Cost Preservation: Let $\mathbf{A} \in \mathbb{R}^{n \times d}$ be a matrix of at least rank $k$ and $\bar{\tau}_i(\mathbf{A})$ be defined as in Eqn. 1. Construct $\mathbf{C}$ following the DRLS algorithm described in Sec. 2.1. Then $\mathbf{C}$ satisfies, for any rank $k$ orthogonal projection $\mathbf{X} \in \mathbb{R}^{n \times n}$,*

$$(1 - \epsilon)||\mathbf{A} - \mathbf{X}\mathbf{A}||_F^2 \leq ||\mathbf{C} - \mathbf{X}\mathbf{C}||_F^2 \leq ||\mathbf{A} - \mathbf{X}\mathbf{A}||_F^2. \tag{9}$$

*To simplify the bookkeeping, we prove the lower bound of Theorem 3 with $(1 - \alpha\epsilon)$ error ($\alpha = 2(2 + \sqrt{2})$), and assume $0 < \epsilon < \frac{1}{2}$.*

Projection-cost preservation bounds were formalized recently in Feldman et al. (2013); Cohen et al. (2015). Bounds of this type are important because it means that low-rank projection problems can be solved with $\mathbf{C}$ instead of $\mathbf{A}$ while maintaining the projection cost. Furthermore, the projection-cost preservation bound has implications for $k$-means clustering, because the $k$-means objective function can be written in terms of the orthogonal rank-$k$ cluster indicator matrix (Boutsidis et al., 2009).[1] Note that our rank-$k$ projection-cost preservation bound holds for an un-weighted column subset of $\mathbf{A}$.

A useful lemma on an approximate ridge leverage score kernel comes from combining Theorem 1 and 3.

**Lemma 1.** *Approximate ridge leverage score kernel: Let $\mathbf{A} \in \mathbb{R}^{n \times d}$ be a matrix of at least rank $k$ and $\bar{\tau}_i(\mathbf{A})$ be defined as in Eqn. 1. Construct $\mathbf{C}$ following the DRLS algorithm described in*

*Sec. 2.1. Let $\alpha$ be the coefficient in the lower bound of Theorem 3 and assume $0 < \epsilon < \frac{1}{2}$. Let $\mathbf{K}(\mathbf{M}) = \left(\mathbf{M}\mathbf{M}^T + \frac{1}{k}||\mathbf{M}_{\setminus k}||_F^2\mathbf{I}\right)^+$ for matrix $\mathbf{M} \in \mathbb{R}^{n \times l}$. Then $\mathbf{K}(\mathbf{C})$ and $\mathbf{K}(\mathbf{A})$ satisfy,*

$$\mathbf{K}(\mathbf{A}) \preceq \mathbf{K}(\mathbf{C}) \preceq \frac{1}{1 - (\alpha + 1)\epsilon}\mathbf{K}(\mathbf{A}). \tag{10}$$

**Theorem 4.** *Approximate Ridge Regression with DRLS: Let $\mathbf{A} \in \mathbb{R}^{n \times d}$ be a matrix of at least rank $k$ and $\bar{\tau}_i(\mathbf{A})$ be defined as in Eqn. 1. Construct $\mathbf{C}$ following the DRLS algorithm described in Sec. 2.1, let $\alpha$ be the coefficient in the lower bound of Theorem 3, and assume $0 < \epsilon < \frac{1}{2\alpha} < \frac{1}{2}$. Choose the regularization parameter $\lambda_2 = \frac{||\mathbf{M}_{\setminus k}||_F^2}{k}$ for ridge regression with a matrix $\mathbf{M}$ (Eqn. 3). Under these conditions, the statistical risk $\mathcal{R}(\hat{\mathbf{y}}_\mathbf{C})$ of the ridge regression estimator $\hat{\mathbf{y}}_\mathbf{C}$ is bounded by the statistical risk $\mathcal{R}(\hat{\mathbf{y}}_\mathbf{A})$ of the ridge regression estimator $\hat{\mathbf{y}}_\mathbf{A}$:*

$$\mathcal{R}(\hat{\mathbf{y}}_\mathbf{C}) \leq (1 + \beta\epsilon)\mathcal{R}(\hat{\mathbf{y}}_\mathbf{A}), \tag{11}$$

*where $\beta = \frac{2\alpha(-1 + 2\alpha + 3\alpha^2)}{(1-\alpha)^2}$.*

Theorem 4 means that there are bounds on the statistical risk for substituting the DRLS selected column subset matrix for the complete matrix when performing ridge regression with the appropriate regularization parameter. Performing ridge regression with the column subset $\mathbf{C}$ effectively forces coefficients to be zero and adds the benefits of automatic feature selection to the $L_2$ regularization problem. We also note that the proof of Theorem 4 relies only on Theorem 1 and Theorem 3 and facts from linear algebra, so a randomized selection of weighted column subsets that obey similar bounds to Theorem 1 and Theorem 3 (e.g. Cohen et al. (2017)) will also have bounded statistical risk, albeit with a different coefficient $\beta$. As a point of comparison, consider the elastic net minimization with our ridge regression regularization parameter:

$$\hat{\mathbf{x}}^E = \underset{\mathbf{x}}{\operatorname{argmin}} \left( ||\mathbf{y} - \mathbf{A}\mathbf{x}||_2^2 + \frac{1}{k}||\mathbf{A}_{\setminus k}||_F^2||\mathbf{x}||_2^2 + \lambda_1 \sum_{j=1}^d |\mathbf{x}_j| \right). \tag{12}$$

The risk of elastic net $\mathcal{R}(\hat{\mathbf{y}}^E)$ has the following bound in terms of the risk of ridge regression $\mathcal{R}(\hat{\mathbf{y}}_\mathbf{A})$:

$$\mathcal{R}(\hat{\mathbf{y}}^E = \mathbf{A}\hat{\mathbf{x}}^E) = \mathcal{R}(\hat{\mathbf{y}}_\mathbf{A}) + \lambda_1^2 \frac{4d||\mathbf{A}||_2^2}{\frac{1}{k^2}||\mathbf{A}_{\setminus k}||_F^4} \tag{13}$$

This comes from a slight re-working of Theorem 3.1 of Zou and Zhang (2009). The bounds for the elastic net risk and $\mathcal{R}(\hat{\mathbf{y}}_\mathbf{C})$ are comparable when $\lambda_1^2 \approx \frac{\beta\epsilon}{k^2}||\mathbf{A}_{\setminus k}||_F^4 \frac{\mathcal{R}(\hat{\mathbf{y}}_\mathbf{A})}{4d||\mathbf{A}||_2^2}$.

Ridge regression is a special case of kernel ridge regression with a linear kernel. While previous work in kernel ridge regression has considered the use of ridge leverage scores to approximate the symmetric kernel matrix by selecting a subset of $n$ informative samples (Alaoui and Mahoney, 2015; Rudi et al., 2015), to our knowledge, no previous work has used ridge leverage scores to approximate the symmetric kernel matrix using ridge leverage scores to select a subset of the $f$ informative features (after the feature mapping of the $d$-dimensional data points). The latter case would be the natural generalization of Theorem 4 to non-linear kernels, and remains an interesting open question. Lastly, we note that placing statistical assumptions on $\mathbf{A}$ in the spirit of (Rudi et al., 2015) may lead to an improved bound for random designs for $\mathbf{A}$.

**Theorem 5.** *Ridge Leverage Power-law Decay: Let $\mathbf{A} \in \mathbb{R}^{n \times d}$ be a matrix of at least rank $k$ and $\bar{\tau}_i(\mathbf{A})$ be defined as in Eqn. 1. Furthermore, let the ridge leverage scores exhibit power-law decay in the sorted column index $\pi_i$,*

$$\bar{\tau}_{\pi_i}(\mathbf{A}) = \pi_i^{-a}\bar{\tau}_{\pi_0}(\mathbf{A}) \qquad a > 1. \tag{14}$$

*Construct $\mathbf{C}$ following the DRLS algorithm described in Sec. 2.1. The number of sample columns selected by DRLS is,*

$$|\Theta| \leq \max\left( \left(\frac{4k}{\epsilon}\right)^{\frac{1}{a}} - 1, \left(\frac{4k}{(a-1)\epsilon}\right)^{\frac{1}{a-1}} - 1, k \right). \tag{15}$$

Theorem 3 of Papailiopoulos et al. (2014) introduces the concept of power-law decay behavior for leverage scores for rank-$k$ subspace leverage scores. Our Theorem 5 is an adaptation of Papailiopoulos et al. (2014)'s Theorem 3 for ridge leverage scores.

An obvious extension of Eqn. 7 is the following bound,

$$(1-\epsilon)\mathbf{A}\mathbf{A}^T - \frac{\epsilon}{k}||\mathbf{A}_{\backslash k}||_F^2\mathbf{I} \preceq \mathbf{C}\mathbf{C}^T \preceq (1+\epsilon)\mathbf{A}\mathbf{A}^T + \frac{\epsilon}{k}||\mathbf{A}_{\backslash k}||_F^2\mathbf{I}, \qquad (16)$$

which also holds for $\mathbf{C}$ selected by ridge leverage random sampling methods with $O(\frac{k}{\epsilon^2}\ln(\frac{k}{\delta}))$ weighted columns and failure probability $\delta$ Cohen et al. (2017). Thus, DRLS selects fewer columns with the same accuracy $\epsilon$ in Eqn. 16 for power-law decay in the ridge leverage scores when,

$$\max\left(\left(\tfrac{4k}{\epsilon}\right)^{\frac{1}{a}} - 1, \left(\tfrac{4k}{(a-1)\epsilon}\right)^{\frac{1}{a-1}} - 1, k\right) < C\frac{k}{\epsilon^2}\ln\left(\tfrac{k}{\delta}\right), \qquad (17)$$

where $C$ is an absolute constant. In particular, when $a \geq 2$, the number of columns deterministically sampled is $\mathcal{O}(k)$.[2]

## 4 Biological Data Illustration

We provide a biological data illustration of ridge leverage scores and ridge regression with multi-omic data from lower-grade glioma (LGG) tumor samples collected by the TCGA Research Network (http://cancergenome.nih.gov/). Diffuse lower-grade gliomas are infiltrative brain tumors that occur most frequently in the cerebral hemisphere of adults.

The data is publicly available and hosted by the Broad Institute's GDAC Firehose (Broad Institute of MIT and Harvard, 2016). We download the data using the $R$ tool $TCGA2STAT$ (Wan et al., 2016). $TCGA2STAT$ imports the latest available version-stamped standardized Level 3 dataset on Firehose. The data collection and data platforms are discussed in detail in the original paper (The Cancer Genome Atlas Research Network, 2015).

We use the following multi-omic data types: mutations ($d = 4845$), DNA copy number (alteration ($d = 22618$) and variation ($d = 22618$)), messenger RNA (mRNA) expression ($d = 20501$), and microRNA expression ($d = 1046$). Methylation data is also available, but we omit it due to memory constraints. The mRNA and microRNA data is normalized. DNA copy number (variation and alteration) has an additional pre-processing step; the segmentation data reported by TCGA is turned into copy number using the $R$ tool $CNtools$ (Zhang, 2015) that is imbedded in $TCGA2STAT$. The mutation data is filtered based on status and variant classification and then aggregated at the gene level (Wan et al., 2016).

There are 280 tumor samples and $d = 71628$ multi-omic features in the downloaded dataset. We are interested in performing ridge regression with the biologically meaningful outcome variables relating to mutations of the "IDH1" and "IDH2" gene and deletions of the "1p/19q" chromosome arms ("codel"). These variables were shown to be predictive of favorable clinical outcomes and can be found in the supplemental tables (The Cancer Genome Atlas Research Network, 2015). We restrict to samples with these outcome variables (275 tumor samples), and we drop an additional sample ("TCGA-CS-4944") because it is an outlier with respect to the $k = 3$ SVD projection of the samples. This leaves a total of 274 tumor samples with outcome variables "IDH" (a mutation in either "IDH1" or "IDH2") and "codel" for the analysis.

Lastly, we drop all multi-omic features that have zero columns and greater than $10\%$ missing data on the 274 tumor samples. We the replace missing values with the mean of the column. This leaves a final multi-omic feature set of $d = 68522$ for the 274 tumor samples. Our final matrix $\mathbf{A} \in \mathbf{R}^{274 \times 68522}$ is column mean-centered. Figure 1 shows a pie chart of the breakdown of the final matrix $\mathbf{A}$'s multi-omic feature types.

### 4.1 Ridge leverage score sampling

Figure 2 shows the spectrum of eigenvalues of $\mathbf{A}\mathbf{A}^T$ for LGG. The eigenvalues range of multiple orders of magnitude. We choose $k = 3$ for the DRLS algorithm because these components are

meaningful for the "IDH" and "codel" outcome variables (see Figures 3, 4 , and 5). The top three components capture $79\%$ of the Frobenius norm $|\mathbf{A}|_F^2$. Applying the DRLS algorithm with $k = 3, \epsilon = 0.1$ leads to $|\Theta| = 1512$, selecting approximately $0.02\%$ of the total multi-omic features for the column subset matrix $\mathbf{C}$. The majority of the features selected are mRNA ($1473$ features), and the remainder are microRNA ($39$ features). Figure 6 shows the relationship between the number of columns kept, $|\Theta|$, and $\tilde{\epsilon} = \sum_{i \notin \Theta} \bar{\tau}_i(\mathbf{A})$ for the $k = 3$ ridge leverage scores. Only a small error penalty is incurred by a dramatic reduction in the number of columns kept according to Algorithm 1. Figure 7 shows the power-law decay of the LGG $k = 3$ ridge leverage scores with sorted column index. This LGG multi-omic data example shows that ridge leverage score power-law decay occurs in the wild. Figure 8 shows a histogram of the ratio of $||\mathbf{C} - \mathbf{XC}||_F^2 / ||\mathbf{A} - \mathbf{XA}||_F^2$ for $1000$ random rank-$k = 3$ orthogonal projections $\mathbf{X}$. The projections are chosen as the first $3$ directions from an orthogonal basis randomly selected with respect to the Haar measure for the orthogonal group $O(n)$ (Mezzadri, 2006). This confirms that the projection cost empirically has very small error. Lastly, Figure 9 illustrates the $k = 3$ ridge leverage score regularization of the classical leverage score for the LGG multi-omic features. As expected, many of the columns' ridge leverage scores exhibit shrinkage when compared to the classical leverage scores. Table 2 includes ratios derived from the full data matrix $\mathbf{A}$ and the column subset matrix $\mathbf{C}$ selected by the DRLS algorithm with $k = 3, \epsilon = 0.1$.

## 4.2 Ridge regression with ridge leverage score sampling

We perform ridge regression with the appropriate regularization parameter for two biologically meaningful outcome variables; the first is whether either the "IDH1" or the "IDH2" gene is mutated and the second whether the "1p/19q" chromosome arms have deletions ("codel"). We encode the status of each event as $\pm 1$. Figures 3, 4 , and 5 show the top three SVD projections for the tumor samples, colored by the combined status for "IDH" and "codel". No tumor samples have the "1p/19q" codeletion and no "IDH" mutation. Visual inspection of the SVD plot confirms that this is a reasonable regression problem for "IDH" and a difficult regression problem for "codel"; also, logisitic regression would be more natural for binary outcomes. We proceed anyway, since our objective is to compare ridge regression with all of the features ($\mathbf{A}$) to ridge regression with the DRLS subset ($\mathbf{C}$) on realistic biological data. Figures 10 and 11 confirm that the ridge regression fits are close ($\hat{\mathbf{y}}_{\mathbf{A}} - \hat{\mathbf{y}}_{\mathbf{C}}$) for all the tumor samples. Figures 12 and 13 confirm that the ridge regression coefficients are close ($\hat{\mathbf{x}}_{\mathbf{A}} - \hat{\mathbf{x}}_{\mathbf{C}}$) for all the tumor samples. Figure 14 and 15 illustrate the overall performance of ridge regression for these two outcome variables.

Lastly, we simulate $274$ samples $\mathbf{y}$ according to the linear model (Eqn. 4), where $\mathbf{y}^* = \mathbf{Ax}^*$, the coefficients $\mathbf{x}^* \sim \mathcal{N}(0, \mathbf{I})$, and $\mathbf{A}$ is the LGG multi-omic feature matrix. We choose $\sigma^2 = \{10^{-3}, 1, 10^3\}$. We perform ridge regression with $\mathbf{A}$ and then again with $\mathbf{C}$ in accordance with Theorem 4. We calculate the risks $\mathcal{R}(\hat{\mathbf{y}}_{\mathbf{A}})$ and $\mathcal{R}(\hat{\mathbf{y}}_{\mathbf{C}})$ and find that Theorem 4 is not violated. Table 2 shows the risk ratios $\mathcal{R}(\hat{\mathbf{y}}_{\mathbf{C}})/\mathcal{R}(\hat{\mathbf{y}}_{\mathbf{A}})$ along with other relevant ratios for the ridge leverage scores.

Table 2: Ridge leverage score ratios for $k = 3, \epsilon = 0.1$ for LGG tumor multi-omic data. The ratios are near one, as expected. Ridge regression risk ratio $\mathcal{R}(\hat{\mathbf{y}}_{\mathbf{C}})/\mathcal{R}(\hat{\mathbf{y}}_{\mathbf{A}})$ for data simulated from the LGG multi-omic matrix $\mathbf{A}$ and Eqn. 4.

| | ave($\boldsymbol{\Sigma}_{\mathbf{C}}^2/\boldsymbol{\Sigma}^2$) | $||\mathbf{C}_{\setminus k}||_F^2/||\mathbf{A}_{\setminus k}||_F^2$ | ave($\bar{\boldsymbol{\Sigma}}^2/\bar{\boldsymbol{\Sigma}}_{\mathbf{C}}^2$) | $\sigma^2$ | $\mathcal{R}(\hat{\mathbf{y}}_{\mathbf{C}})/\mathcal{R}(\hat{\mathbf{y}}_{\mathbf{A}})$ |
|---|---|---|---|---|---|
| Algorithm 1 | 0.85 | 0.97 | 1.03 | $10^{-3}$ | 0.99 |
| $k = 3, \epsilon = 0.1$ | | | | $10^0$ | 0.99 |
| | | | | $10^3$ | 0.99 |

## Acknowledgements

Research reported in this publication was supported by the National Human Genome Research Institute of the National Institutes of Health under Award Number F32HG008713. The content is solely the responsibility of the authors and does not necessarily represent the official views of the National Institutes of Health. SRM thanks Michael Mahoney, Ahmed El Alaoui, Elaine Angelino, and Kai Rothauge for thoughtful comments and the Barcellos and Pachter Labs.

## Supporting Information

Software in the form of python and R code is available at `https://github.com/srmcc/deterministic-ridge-leverage-sampling`. Code for downloading the data and reproducing all of the figures is included. Proofs and figures are included in the Supplementary Material.

## Footnotes

[1]Thanks to Michael Mahoney for this point.

[2]Thanks to Ahmed El Alaoui for this point.

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
