[Supplementary Material · det_ridge_leverage_NIPS_SM.pdf]

# Supplementary Material: Ridge Regression and Provable Deterministic Ridge Leverage Score Sampling

**Shannon R. McCurdy**
California Institute for Quantitative Biosciences
UC Berkeley
Berkeley, CA 94702
smccurdy@berkeley.edu

## 1 Proofs

### 1.1 Brief Review

See Horn and Johnson (2013) for a linear algebra review. A square complex matrix $\mathbf{F}$ is *Hermitian* if $\mathbf{F} = \mathbf{F}^\dagger$. Symmetric positive semi-definite (SPSD) matrices are Hermitian matrices. The set of $n \times n$ Hermitian matrices is a real linear space. As such, it is possible to define a *partial ordering* (also called a Loewner partial ordering, denoted by $\preceq$) on the real linear space. One matrix is "greater" than another if their difference lies in the closed convex cone of SPSD matrices. Let $\mathbf{F}, \mathbf{G}$ be Hermitian and the same size, and $\mathbf{x}$ a complex vector of compatible dimension. Then,

$$\mathbf{F} \preceq \mathbf{G} \iff \mathbf{x}^\dagger \mathbf{F} \mathbf{x} \le \mathbf{x}^\dagger \mathbf{G} \mathbf{x} \quad \forall \mathbf{x} \ne \mathbf{0}. \tag{SM1}$$

If $\mathbf{F}$ is Hermitian with smallest and largest eigenvalues $\lambda_{\min}(\mathbf{F}), \lambda_{\max}(\mathbf{F})$, respectively, then,

$$\lambda_{\min}(\mathbf{F})\mathbf{I} \preceq \mathbf{F} \preceq \lambda_{\max}(\mathbf{F})\mathbf{I}. \tag{SM2}$$

Let $\mathbf{F}, \mathbf{G}$ be Hermitian and the same size, and let $\mathbf{H}$ be any complex rectangular matrix of compatible dimension. The *conjugation rule* is,

$$\text{If } \mathbf{F} \preceq \mathbf{G}, \text{ then } \mathbf{H}\mathbf{F}\mathbf{H}^\dagger \preceq \mathbf{H}\mathbf{G}\mathbf{H}^\dagger. \tag{SM3}$$

In addition, let $\lambda_i(\mathbf{F})$ and $\lambda_i(\mathbf{G})$ be the non-decreasingly ordered eigenvalues of $\mathbf{F}, \mathbf{G}$. Then,

$$\text{If } \mathbf{F} \preceq \mathbf{G}, \text{ then } \forall i, \ \lambda_i(\mathbf{F}) \le \lambda_i(\mathbf{G}). \tag{SM4}$$

Since the trace of a matrix $\mathbf{F}$ is the sum of its eigenvalues, $\operatorname{tr}\mathbf{F} = \sum_i \lambda_i(\mathbf{F})$, and the Loewner ordering implies the ordering of eigenvalues (Eqn. SM4), the Loewner ordering also implies the ordering of their sum,

$$\text{If } \mathbf{F} \preceq \mathbf{G}, \text{ then } \operatorname{tr}\mathbf{F} \le \operatorname{tr}\mathbf{G}. \tag{SM5}$$

### 1.2 Proof of ridge leverage score sum

The sum of ridge leverage scores is,

$$
\begin{aligned}
\sum_{i=1}^d \bar{\tau}_i(\mathbf{A}) &= \operatorname{tr}\mathbf{A}^T \left( \mathbf{A}\mathbf{A}^T + \frac{1}{k}||\mathbf{A} - \mathbf{A}_k||_F^2 \mathbf{I} \right)^+ \mathbf{A} \\
&= \operatorname{tr}\mathbf{A}^T \mathbf{U}\bar{\mathbf{\Sigma}}^{-2}\mathbf{U}^T\mathbf{A} = \operatorname{tr}\bar{\mathbf{\Sigma}}^{-2}\mathbf{\Sigma}^2,
\end{aligned}
\tag{SM6}
$$

where $\bar{\boldsymbol{\Sigma}}$ is diagonal and $\bar{\boldsymbol{\Sigma}}_{i,i}^2 = \boldsymbol{\Sigma}_{i,i}^2 + \frac{1}{k}||\mathbf{A}_{\backslash k}||_F^2$. We split the sum into two parts,

$$= \sum_{i=1}^{k} \frac{\boldsymbol{\Sigma}_{i,i}^2}{\boldsymbol{\Sigma}_{i,i}^2 + \frac{1}{k}||\mathbf{A}_{\backslash k}||_F^2} + \sum_{i=k+1}^{n} \frac{\boldsymbol{\Sigma}_{i,i}^2}{\boldsymbol{\Sigma}_{i,i}^2 + \frac{1}{k}||\mathbf{A}_{\backslash k}||_F^2}$$

$$\leq \sum_{i=1}^{k} \frac{\boldsymbol{\Sigma}_{i,i}^2}{\boldsymbol{\Sigma}_{i,i}^2} + \sum_{i=k+1}^{n} \frac{\boldsymbol{\Sigma}_{i,i}^2}{\frac{1}{k}||\mathbf{A}_{\backslash k}||_F^2} = k + k. \tag{SM7}$$

This proof is due to Cohen et al. (2017).

## 1.3 Proof of Theorem 1

The upper bound in Eqn. 7 in Theorem 1 follows from the fact that $\mathbb{0} \preceq \mathbf{I} - \mathbf{SS}^T$ and the conjugation rule (Eqn. SM3),

$$\mathbb{0} \quad \preceq \quad \mathbf{A}(\mathbf{I} - \mathbf{SS}^T)\mathbf{A}^T = \mathbf{AA}^T - \mathbf{CC}^T. \tag{SM8}$$

This upper bound is true for any column selection of $\mathbf{A}$.

For the lower bound in Eqn. 7, consider the quantity,

$$\mathbf{Y} = \bar{\boldsymbol{\Sigma}}^{-1}\mathbf{U}^T\mathbf{A}(\mathbf{I} - \mathbf{SS}^T)\mathbf{A}^T\mathbf{U}\bar{\boldsymbol{\Sigma}}^{-1} = \bar{\boldsymbol{\Sigma}}^{-1}\boldsymbol{\Sigma}\mathbf{V}^T(\mathbf{I} - \mathbf{SS}^T)\mathbf{V}\boldsymbol{\Sigma}\bar{\boldsymbol{\Sigma}}^{-1}.$$

By the conjugation rule (Eqn. SM3) on Eqn. SM8, $\mathbb{0} \preceq \mathbf{Y}$, so $\mathbf{Y}$ is S.P.S.D. By the construction of DRLS (Eqn. 6) $\operatorname{tr}\mathbf{Y} = \sum_{i \notin \Theta}\sum_{l=1}^{n} \bar{\boldsymbol{\Sigma}}_{l,l}^{-2}\boldsymbol{\Sigma}_{l,l}^2 V_{il}^2 = \tilde{\epsilon} < \epsilon$, and because $\mathbf{Y}$ is S.P.S.D., $\lambda_{\max}(\mathbf{Y}) \leq \operatorname{tr}\mathbf{Y}$. By Eqn. SM2 and the previous facts, $\mathbf{Y} \preceq \lambda_{\max}(\mathbf{Y})\mathbf{I} \preceq \epsilon\mathbf{I}$. As a result of the conjugation rule applied to this upper bound,

$$\mathbf{U}\bar{\boldsymbol{\Sigma}}\mathbf{Y}\bar{\boldsymbol{\Sigma}}\mathbf{U}^T = \mathbf{AA}^T - \mathbf{CC}^T \quad \preceq \quad \epsilon\mathbf{U}\bar{\boldsymbol{\Sigma}}^2\mathbf{U}^T = \epsilon\left(\mathbf{AA}^T + \frac{1}{k}||\mathbf{A}_{\backslash k}||_F^2\mathbf{I}\right),$$

and rearrangement leads to the lower bound of Eqn. 7.

## 1.4 Proof of Theorem 2

To prove Eqn. 8, we will use the following lemma.

**Lemma 1.** *(Boutsidis et al. (2011) Lemma 3.1, Eqn. 3.2, specialized to the Frobenius norm): Consider $\mathbf{A} = \mathbf{AZZ}^T + \mathbf{E} \in \mathbb{R}^{n \times d}$, where $\mathbf{Z} \in \mathbb{R}^{n \times k}$ and $\mathbf{Z}^T\mathbf{Z} = \mathbf{I}_k$. Let $\mathbf{S} \in \mathbb{R}^{n \times |\Theta|}(k \leq |\Theta|)$ be any matrix such that $\operatorname{rank}(\mathbf{Z}^T\mathbf{S}) = k$, and let $\mathbf{C} = \mathbf{AS}$. Then,*

$$||\mathbf{A} - \mathbf{CC}^+ A||_F^2 \leq ||\mathbf{A} - \boldsymbol{\Pi}_{\mathbf{C},k}^F(\mathbf{A})||_F^2 \leq ||\mathbf{E}||_F^2\,||\mathbf{S}(\mathbf{Z}^T\mathbf{S})^+||_2^2, \tag{SM9}$$

*where $\boldsymbol{\Pi}_{\mathbf{C},k}^F(\mathbf{A}) = (\mathbf{CC}^+\mathbf{A})_k$ is the best rank-k approximation to $\mathbf{A}$ in the column space of $\mathbf{C}$ with the respect to the Frobenius norm.*

We'll use a slightly relaxed version of Lemma 1,

$$||\mathbf{A} - \mathbf{CC}^+ A||_F^2 \leq ||\mathbf{A} - \boldsymbol{\Pi}_{\mathbf{C},k}^F(\mathbf{A})||_F^2 \leq ||\mathbf{E}||_F^2\,||\mathbf{S}||_2^2||(\mathbf{Z}^T\mathbf{S})^+||_2^2. \tag{SM10}$$

Choosing $\mathbf{Z} = \mathbf{V}_k$, and noting that $||\mathbf{S}||_2^2 = 1$, we have

$$||\mathbf{A} - \mathbf{CC}^+ A||_F^2 \leq ||\mathbf{A} - \boldsymbol{\Pi}_{\mathbf{C},k}^F(\mathbf{A})||_F^2 \leq ||\mathbf{A}_{\backslash k}||_F^2\,||(\mathbf{V}_k^T\mathbf{S})^+||_2^2. \tag{SM11}$$

It remains to calculate $||(\mathbf{V}_k^T\mathbf{S})^+||_2^2 = \sigma_k^{-2}(\mathbf{V}_k\mathbf{SS}^T\mathbf{V}_k^T)$, where $k \leq \operatorname{rank}(\mathbf{A}) = r$. As a consequence of the conjugation rule (Eqn. SM3) applied to Eqn. 7 with pre- (post-)multiplication by $\boldsymbol{\Sigma}_k^{-1}\mathbf{U}_k^T$ ( $\mathbf{U}_k\boldsymbol{\Sigma}_k^{-1}$), we have,

$$(1 - \epsilon)\mathbf{I}_k - \frac{\epsilon}{k}||\mathbf{A}_{\backslash k}||_F^2\boldsymbol{\Sigma}_k^{-2} \quad \preceq \quad \mathbf{V}_k\mathbf{SS}^T\mathbf{V}_k^T \tag{SM12}$$

From Eqn. SM12, the $k^{th}$ eigenvalue $\sigma_k^2(\mathbf{V}_r\mathbf{SS}^T\mathbf{V}_r^T)$ obeys,

$$(1 - 2\epsilon) \leq \sigma_k^2(\mathbf{V}_k\mathbf{SS}^T\mathbf{V}_k^T), \tag{SM13}$$

after using the fact that $\frac{1}{k}||\mathbf{A}_{\backslash k}||_F^2\boldsymbol{\Sigma}_k^{-2} \leq 1$ (by definition). Eqn. SM13 shows that, for $0 < \epsilon < \frac{1}{2}$, $\operatorname{rank}(\mathbf{V}_k^T\mathbf{S}) = k$. Combining Eqn. SM11 and Eqn. SM13 gives Eqn. 8. This proof illustrates the power of the spectral bound (Eqn. 7), since the column subset selection bound (Eqn. 8) is a direct consequence of Eqn. 7.

## 1.5 Proof of Theorem 3

It will be convenient to define the projection matrix $\mathbf{Y} = \mathbf{I} - \mathbf{X}$. Then the upper bound in Eqn. 9 follows directly from upper spectral bound (Eqn.SM8), the conjugation rule (Eqn. SM3), and the trace rule (Eqn. SM5).

The lower projection-cost preservation bound is considerably more involved to prove, and also relies primarily on the spectral bound (Eqn. 7) and facts from linear algebra. Our proof is nearly identical to the proof of Cohen et al. (2017)'s Theorem 4, with only one large deviation and several small differences in constants. We include the full proof for completeness, and point out the major difference.

We split $\mathbf{AA}^T$ and $\mathbf{CC}^T$ into their projections on the top $m$ "head" singular vectors and the remaining "tail" singular vectors. Choose $m$ such that $\Sigma_{m,m}$ is the smallest singular value that obeys $\Sigma_{m,m}^2 \geq \frac{1}{k}||\mathbf{A}_{\backslash k}||_F^2$. Let $\mathbf{P}_m = \mathbf{U}_m\mathbf{U}_m^T$ and $\mathbf{P}_{\backslash m} = \mathbf{U}_{\backslash m}\mathbf{U}_{\backslash m}^T$ be projection matrices. Note that,

$$\text{tr}\left(\mathbf{Y}\mathbf{A}\mathbf{A}^T\mathbf{Y}\right) = \text{tr}\left(\mathbf{Y}\mathbf{A}_m\mathbf{A}_m^T\mathbf{Y}\right) + \text{tr}\left(\mathbf{Y}\mathbf{A}_{\backslash m}\mathbf{A}_{\backslash m}^T\mathbf{Y}\right)$$
$$\text{tr}\left(\mathbf{Y}\mathbf{C}\mathbf{C}^T\mathbf{Y}\right) = \text{tr}\left(\mathbf{Y}\mathbf{P}_m\mathbf{C}\mathbf{C}^T\mathbf{P}_m^T\mathbf{Y}\right) + \text{tr}\left(\mathbf{Y}\mathbf{P}_{\backslash m}\mathbf{C}\mathbf{C}^T\mathbf{P}_{\backslash m}^T\mathbf{Y}\right)$$
$$+ 2\text{tr}\left(\mathbf{Y}\mathbf{P}_m\mathbf{C}\mathbf{C}^T\mathbf{P}_{\backslash m}^T\mathbf{Y}\right). \tag{SM14}$$

### 1.5.1 Head terms

First we bound the terms involving only $\mathbf{P}_m$. Consider Eqn. 7 and the vector $\mathbf{y} = \mathbf{P}_m\mathbf{x}$ for any vector $\mathbf{x}$. Eqn. 7 gives,

$$(1-\epsilon)\mathbf{x}^T\mathbf{P}_m\mathbf{A}\mathbf{A}^T\mathbf{P}_m\mathbf{x} - \frac{\epsilon}{k}||\mathbf{A}_{\backslash k}||_F^2\mathbf{y}^T\mathbf{y} \leq \mathbf{y}^T\mathbf{C}\mathbf{C}^T\mathbf{y}$$
$$(1-2\epsilon)\mathbf{x}^T\mathbf{P}_m\mathbf{A}\mathbf{A}^T\mathbf{P}_m\mathbf{x} \leq \mathbf{x}^T\mathbf{P}_m\mathbf{C}\mathbf{C}^T\mathbf{P}_m\mathbf{x},$$

because by the artful definition of $m$, $\mathbf{x}^T\mathbf{P}_m\mathbf{A}\mathbf{A}^T\mathbf{P}_m\mathbf{x} \geq \frac{1}{k}||\mathbf{A}_{\backslash k}||_F^2\mathbf{y}^T\mathbf{y}$. Therefore we have,

$$(1-2\epsilon)\mathbf{P}_m\mathbf{A}\mathbf{A}^T\mathbf{P}_m \preceq \mathbf{P}_m\mathbf{C}\mathbf{C}^T\mathbf{P}_m, \tag{SM15}$$

and finally,

$$(1-2\epsilon)\text{tr}\left(\mathbf{Y}\mathbf{A}_m\mathbf{A}_m^T\mathbf{Y}\right) \leq \text{tr}\left(\mathbf{Y}\mathbf{P}_m\mathbf{C}\mathbf{C}^T\mathbf{P}_m\mathbf{Y}\right). \tag{SM16}$$

### 1.5.2 Tail terms

To bound the lower singular directions of $\mathbf{A}$, we decompose $\text{tr}\left(\mathbf{Y}\mathbf{A}_{\backslash m}\mathbf{A}_{\backslash m}^T\mathbf{Y}\right)$ further as,

$$\text{tr}\left(\mathbf{Y}\mathbf{A}_{\backslash m}\mathbf{A}_{\backslash m}^T\mathbf{Y}\right) = \text{tr}\left(\mathbf{A}_{\backslash m}\mathbf{A}_{\backslash m}^T\right) - \text{tr}\left(\mathbf{X}\mathbf{A}_{\backslash m}\mathbf{A}_{\backslash m}^T\mathbf{X}\right), \tag{SM17}$$

and analogously for $\mathbf{C}$.

The upper spectral bound (Eqn.SM8), the conjugation rule (Eqn. SM3) gives,

$$\mathbf{P}_{\backslash m}\mathbf{C}\mathbf{C}^T\mathbf{P}_{\backslash m} \preceq \mathbf{P}_{\backslash m}\mathbf{A}\mathbf{A}^T\mathbf{P}_{\backslash m}, \tag{SM18}$$

The conjugation rule (Eqn. SM3), and the trace rule (Eqn. SM5) give,

$$\text{tr}\left(\mathbf{X}\mathbf{P}_{\backslash m}\mathbf{C}\mathbf{C}^T\mathbf{P}_{\backslash m}\mathbf{X}\right) \leq \text{tr}\left(\mathbf{X}\mathbf{P}_{\backslash m}\mathbf{A}\mathbf{A}^T\mathbf{P}_{\backslash m}\mathbf{X}\right). \tag{SM19}$$

Next we consider $||\mathbf{P}_{\backslash m}\mathbf{A}||_F^2 - ||\mathbf{P}_{\backslash m}\mathbf{C}||_F^2$. In Cohen et al. (2017)'s proof, a scalar Chernoff bound is used for $||\mathbf{P}_{\backslash m}\mathbf{A}||_F^2 - ||\mathbf{P}_{\backslash m}\mathbf{C}||_F^2$ (Cohen et al. (2017) Section 4.3, Eqn. 17). Since our matrix $\mathbf{C}$ is constructed deterministically, we will prove and substitute the following bound (Eqn. SM20),

$$0 \leq ||\mathbf{P}_{\backslash m}\mathbf{A}||_F^2 - ||\mathbf{P}_{\backslash m}\mathbf{C}||_F^2 \leq 2\epsilon||\mathbf{A}_{\backslash k}||_F^2. \tag{SM20}$$

for the scalar Chernoff bound.

To prove Eqn. SM20, we first note the lower bound follows directly from upper spectral bound (Eqn. SM8), the conjugation rule (Eqn. SM3), and the trace rule (Eqn. SM5). To prove the upper bound, we rewrite the difference in Frobenius norms as a difference in sums over column norms:

$$
\begin{aligned}
||\mathbf{P}_{\backslash m}\mathbf{A}||_F^2 \quad - \quad ||\mathbf{P}_{\backslash m}\mathbf{C}||_F^2 &= \sum_{j \notin \Theta} ||\mathbf{P}_{\backslash m}\mathbf{a}_j||_2^2 = \sum_{j \notin \Theta} \operatorname{tr} \mathbf{a}_j^T \mathbf{P}_{\backslash m}\mathbf{P}_{\backslash m}\mathbf{a}_j \\
&\leq \quad \frac{2}{k}||\mathbf{A}_{\backslash k}||_F^2 \sum_{j \notin \Theta} \operatorname{tr} \mathbf{a}_j^T \mathbf{P}_{\backslash m} \mathbf{U}_{\backslash m} \bar{\boldsymbol{\Sigma}}^{-2} \mathbf{U}_{\backslash m}^T \mathbf{P}_{\backslash m}\mathbf{a}_j \\
&\leq \quad \frac{2}{k}||\mathbf{A}_{\backslash k}||_F^2 \sum_{j \notin \Theta} \operatorname{tr} \mathbf{a}_j^T \mathbf{U} \bar{\boldsymbol{\Sigma}}^{-2} \mathbf{U}\mathbf{a}_j \\
&= \quad \frac{2\tilde{\epsilon}}{k}||\mathbf{A}_{\backslash k}||_F^2 \leq 2\epsilon||\mathbf{A}_{\backslash k}||_F^2. \quad\quad \text{(SM21)}
\end{aligned}
$$

The first inequality follows from the fact that $\bar{\boldsymbol{\Sigma}}_{i,i}^2 = \boldsymbol{\Sigma}_{i,i}^2 + \frac{1}{k}||\mathbf{A}_{\backslash k}||_F^2 \leq \frac{2}{k}||\mathbf{A}_{\backslash k}||_F^2$ for $i \geq m$. As a result, $\mathbf{P}_{\backslash m} \preceq \frac{2}{k}||\mathbf{A}_{\backslash k}||_F^2 \mathbf{U}_{\backslash m} \bar{\boldsymbol{\Sigma}}^{-2} \mathbf{U}_{\backslash m}^T$.

Combining the upper bound of Eqn. SM20 with Eqn. SM19 gives,

$$
\operatorname{tr}\left(\mathbf{Y}\mathbf{A}_{\backslash m}\mathbf{A}_{\backslash m}^T\mathbf{Y}\right) - 2\epsilon||\mathbf{A}_{\backslash k}||_F^2 \leq \operatorname{tr}\left(\mathbf{Y}\mathbf{P}_{\backslash m}\mathbf{C}\mathbf{C}^T\mathbf{P}_{\backslash m}\mathbf{Y}\right). \quad\quad \text{(SM22)}
$$

### 1.5.3   Cross terms

Finally, we show $\operatorname{tr}\left(\mathbf{Y}\mathbf{P}_m\mathbf{C}\mathbf{C}^T\mathbf{P}_{\backslash m}\mathbf{Y}\right)$ is small. We rewrite it as,

$$
\operatorname{tr}\left(\mathbf{Y}\mathbf{P}_m\mathbf{C}\mathbf{C}^T\mathbf{P}_{\backslash m}\mathbf{Y}\right) = \operatorname{tr}\left(\mathbf{Y}\mathbf{A}\mathbf{A}^T(\mathbf{A}\mathbf{A}^T)^+\mathbf{P}_m\mathbf{C}\mathbf{C}^T\mathbf{P}_{\backslash m}\right), \quad\quad \text{(SM23)}
$$

using the cyclic property of the trace and the fact that $\mathbf{P}_m\mathbf{C}\mathbf{C}^T\mathbf{P}_{\backslash m}^T$ is in $\mathbf{A}$'s column span. Because $(\mathbf{A}\mathbf{A}^T)^+$ is semi-positive definite and defines a semi-inner product, we can use the Cauchy-Schwarz inequality,

$$
\begin{aligned}
|\operatorname{tr}\left(\mathbf{Y}\mathbf{A}\mathbf{A}^T(\mathbf{A}\mathbf{A}^T)^+\mathbf{P}_m\mathbf{C}\mathbf{C}^T\mathbf{P}_{\backslash m}^T\right)| &\leq \left(\operatorname{tr}\left(\mathbf{Y}\mathbf{A}\mathbf{A}^T(\mathbf{A}\mathbf{A}^T)^+\mathbf{A}\mathbf{A}^T\mathbf{Y}\right)\right)^{\frac{1}{2}} \\
&\quad \times \left(\operatorname{tr}\left(\mathbf{P}_{\backslash m}\mathbf{C}\mathbf{C}^T\mathbf{P}_m^T(\mathbf{A}\mathbf{A}^T)^+\mathbf{P}_m\mathbf{C}\mathbf{C}^T\mathbf{P}_{\backslash m}^T\right)\right)^{\frac{1}{2}} \\
&= \left(\operatorname{tr}\left(\mathbf{Y}\mathbf{A}\mathbf{A}^T\mathbf{Y}\right)\right)^{\frac{1}{2}} ||\mathbf{P}_{\backslash m}\mathbf{C}\mathbf{C}^T\mathbf{U}_m\boldsymbol{\Sigma}_m^{-1}||_F.
\end{aligned}
$$
$$\text{(SM24)}$$

The square of the second term decomposes as,

$$
||\mathbf{P}_{\backslash m}\mathbf{C}\mathbf{C}^T\mathbf{U}_m\boldsymbol{\Sigma}_m^{-1}||_F^2 = \sum_{i=1}^m ||\mathbf{P}_{\backslash m}\mathbf{C}\mathbf{C}^T\mathbf{u}_i||_2^2 \boldsymbol{\Sigma}_{i,i}^{-2}, \quad\quad \text{(SM25)}
$$

which is small for every $i$. To show this, we define two convenient vectors. The first is the unit vector $\mathbf{p}_i = \frac{1}{||\mathbf{P}_{\backslash m}\mathbf{C}\mathbf{C}^T\mathbf{u}_i||_2}\mathbf{P}_{\backslash m}\mathbf{C}\mathbf{C}^T\mathbf{u}_i$. Note that $\mathbf{p}_i^T\mathbf{u}_i = 0$. This is convenient because $(\mathbf{p}_i^T\mathbf{C}\mathbf{C}^T\mathbf{u}_i)^2 = ||\mathbf{P}_{\backslash m}\mathbf{C}\mathbf{C}^T\mathbf{u}_i||_2^2$. The second vector is $\mathbf{m} = \boldsymbol{\Sigma}_{i,i}^{-1}\mathbf{u}_i + \frac{\sqrt{k}}{||\mathbf{A}_{\backslash k}||_F}\mathbf{p}_i$. From Eqn. SM8,

$$
\begin{aligned}
\mathbf{m}^T\mathbf{C}\mathbf{C}^T\mathbf{m} &\leq \mathbf{m}^T\mathbf{A}\mathbf{A}^T\mathbf{m} \\
\boldsymbol{\Sigma}_{i,i}^{-2}\mathbf{u}_i^T\mathbf{C}\mathbf{C}^T\mathbf{u}_i + \frac{k}{||\mathbf{A}_{\backslash k}||_F^2}\mathbf{p}_i^T\mathbf{C}\mathbf{C}^T\mathbf{p}_i + \frac{2\sqrt{k}}{\boldsymbol{\Sigma}_{i,i}||\mathbf{A}_{\backslash k}||_F}\mathbf{p}_i^T\mathbf{C}\mathbf{C}^T\mathbf{u}_i &\leq \\
\boldsymbol{\Sigma}_{i,i}^{-2}\mathbf{u}_i^T\mathbf{A}\mathbf{A}^T\mathbf{u}_i + \frac{k}{||\mathbf{A}_{\backslash k}||_F^2}\mathbf{p}_i^T\mathbf{A}\mathbf{A}^T\mathbf{p}_i &= 1 + \frac{k}{||\mathbf{A}_{\backslash k}||_F^2}\mathbf{p}_i^T\mathbf{A}\mathbf{A}^T\mathbf{p}_i.
\end{aligned}
$$
$$\text{(SM26)}$$

From Eqn. SM15, we have,

$$
(1 - 2\epsilon)\boldsymbol{\Sigma}_{i,i} = (1 - 2\epsilon)\mathbf{u}_i\mathbf{A}\mathbf{A}^T\mathbf{u}_i \quad \leq \quad \mathbf{u}_i\mathbf{C}\mathbf{C}^T\mathbf{u}_i. \quad\quad \text{(SM27)}
$$

From Eqn. SM18, we have,

$$\mathbf{p}_i \mathbf{CC}^T \mathbf{p}_i \quad \leq \quad \mathbf{p}_i \mathbf{AA}^T \mathbf{p}_i. \tag{SM28}$$

Using these facts, Eqn. SM26 becomes,

$$(1 - 2\epsilon) + \frac{k}{||\mathbf{A}_{\backslash k}||_F^2} \mathbf{p}_i^T \mathbf{CC}^T \mathbf{p}_i + \frac{2\sqrt{k}}{\mathbf{\Sigma}_{i,i}||\mathbf{A}_{\backslash k}||_F} \mathbf{p}_i^T \mathbf{CC}^T \mathbf{u}_i \leq$$

$$1 + \frac{k}{||\mathbf{A}_{\backslash k}||_F^2} \mathbf{p}_i^T \mathbf{CC}^T \mathbf{p}_i$$

$$(\mathbf{p}_i^T \mathbf{CC}^T \mathbf{u}_i)^2 \leq \epsilon^2 \frac{\mathbf{\Sigma}_{i,i}^2 ||\mathbf{A}_{\backslash k}||_F^2}{k}. \tag{SM29}$$

Returning to Eqn. SM25 with this, we have

$$||\mathbf{P}_{\backslash m} \mathbf{CC}^T \mathbf{U}_m \mathbf{\Sigma}_m^{-1}||_F^2 \leq \sum_{i=1}^m \epsilon^2 \frac{||\mathbf{A}_{\backslash k}||_F^2}{k} \leq 2\epsilon^2 ||\mathbf{A}_{\backslash k}||_F^2. \tag{SM30}$$

The factor of 2 comes from the fact that $m \leq 2k$. After recalling that the Eckart-Young-Mirsky theorem (Eckart and Young, 1936) gives $||\mathbf{A}_{\backslash k}||_F^2 \leq \text{tr}\left(\mathbf{YAA}^T\mathbf{Y}\right)$, we have for Eqn. SM24,

$$|\text{tr}\left(\mathbf{YAA}^T(\mathbf{AA}^T)^+\mathbf{P}_m\mathbf{CC}^T\mathbf{P}_{\backslash m}^T\right)| \leq \sqrt{2}\epsilon\text{tr}\left(\mathbf{YAA}^T\mathbf{Y}\right). \tag{SM31}$$

Combining Eqn. SM14 , Eqn. SM16 , Eqn. SM22, Eqn. SM31 leads to,

$$(1 - 2\epsilon)\text{tr}\left(\mathbf{YA}_m\mathbf{A}_m^T\mathbf{Y}\right) + \text{tr}\left(\mathbf{YA}_{\backslash m}\mathbf{A}_{\backslash m}^T\mathbf{Y}\right) - 2\epsilon||\mathbf{A}_{\backslash k}||_F^2$$

$$-2\sqrt{2}\epsilon\text{tr}\left(\mathbf{YAA}^T\mathbf{Y}\right) \leq \text{tr}\left(\mathbf{YCC}^T\mathbf{Y}\right). \tag{SM32}$$

Again applying $||\mathbf{A}_{\backslash k}||_F^2 \leq \text{tr}\left(\mathbf{YAA}^T\mathbf{Y}\right)$ and subtracting an extra $0 \leq 2\epsilon\text{tr}\left(\mathbf{YA}_{\backslash m}\mathbf{A}_{\backslash m}^T\mathbf{Y}\right)$ from the lefthand side gives,

$$(1 - 2(2 + \sqrt{2})\epsilon)\text{tr}\left(\mathbf{YAA}^T\mathbf{Y}\right) \leq \text{tr}\left(\mathbf{YCC}^T\mathbf{Y}\right), \tag{SM33}$$

proving Theorem 3.

### 1.6 Proof of Lemma 1

Let the SVD of $\mathbf{C}$ be $\mathbf{C} = \mathbf{W}\mathbf{\Sigma}_C\mathbf{Z}^T$. Set $\mathbf{X} = \mathbf{W}_k\mathbf{W}_k^T$. From the left-hand side of Eqn. 9 and the Eckart-Young-Mirsky theorem (Eckart and Young, 1936),

$$(1 - \alpha\epsilon)||\mathbf{A}_{\backslash k}||_F^2 \leq (1 - \alpha\epsilon)||\mathbf{A} - \mathbf{XA}||_F^2 \leq ||\mathbf{C}_{\backslash k}||_F^2. \tag{SM34}$$

Similarly, set $\mathbf{X} = \mathbf{U}_k\mathbf{U}_k^T$. From the right-hand side of Eqn. 9,

$$||\mathbf{C}_{\backslash k}||_F^2 \leq ||\mathbf{C} - \mathbf{XC}||_F^2 \leq ||\mathbf{A}_{\backslash k}||_F^2. \tag{SM35}$$

This means that,

$$(1 - \alpha\epsilon)||\mathbf{A}_{\backslash k}||_F^2\mathbf{I} \preceq ||\mathbf{C}_{\backslash k}||_F^2\mathbf{I} \preceq ||\mathbf{A}_{\backslash k}||_F^2\mathbf{I}. \tag{SM36}$$

Adding $1/k$ times Eqn. SM36 to Eqn. 7 gives,

$$(1 - \epsilon)\mathbf{K}(\mathbf{A})^{-1} - \frac{\alpha\epsilon}{k}||\mathbf{A}_{\backslash k}||_F^2\mathbf{I} \quad \preceq \quad \mathbf{K}(\mathbf{C})^{-1} \preceq \mathbf{K}(\mathbf{A})^{-1}. \tag{SM37}$$

Noting that we can subtract $\alpha\epsilon\mathbf{AA}^T$ from the left-most side of Eqn. SM37 and that all of the matrices are invertible gives the lemma.

## 1.7 Proof of Theorem 4

For ridge regression with matrix $\mathbf{A}$ and estimator $\hat{\mathbf{y}} = \mathbf{A}\hat{\mathbf{x}}$, the statistical risk of the estimator $\mathcal{R}(\hat{\mathbf{y}}_{\mathbf{A}})$ is,

$$\mathcal{R}(\hat{\mathbf{y}}_{\mathbf{A}}) = \tfrac{1}{n}\mathbb{E}_{\boldsymbol{\xi}}\left[\|\mathbf{A}\left(\mathbf{A}^T\mathbf{A} + \tfrac{\|\mathbf{A}_{\setminus k}\|_F^2}{k}\mathbf{I}\right)^{-1}\mathbf{A}^T(\mathbf{y}^* + \sigma^2\boldsymbol{\xi}) - \mathbf{y}^*\|_2^2\right]. \tag{SM38}$$

Decomposing into bias and variance terms, taking the expectation, and using the Woodbury matrix inversion formula gives,

$$
\begin{aligned}
\mathcal{R}(\hat{\mathbf{y}}_{\mathbf{A}}) &= \tfrac{1}{n}\|\left(\mathbf{A}\left(\mathbf{A}^T\mathbf{A} + \tfrac{\|\mathbf{A}_{\setminus k}\|_F^2}{k}\mathbf{I}\right)^{-1}\mathbf{A}^T - \mathbf{I}\right)\mathbf{y}^*\|_2^2 \\
&+ \tfrac{\sigma^2}{n}\mathrm{tr}\left(\left(\mathbf{A}\left(\mathbf{A}^T\mathbf{A} + \tfrac{\|\mathbf{A}_{\setminus k}\|_F^2}{k}\mathbf{I}\right)^{-1}\mathbf{A}^T\right)^2\right) \\
&= \tfrac{\|\mathbf{A}_{\setminus k}\|_F^4}{nk^2}\|\left(\mathbf{A}\mathbf{A}^T + \tfrac{\|\mathbf{A}_{\setminus k}\|_F^2}{k}\right)^{-1}\mathbf{y}^*\|_2^2 \\
&+ \tfrac{\sigma^2}{n}\mathrm{tr}\left(\left(\mathbf{A}\left(\mathbf{A}^T\mathbf{A} + \tfrac{\|\mathbf{A}_{\setminus k}\|_F^2}{k}\mathbf{I}\right)^{-1}\mathbf{A}^T\right)^2\right) \\
&\equiv \mathrm{bias}(\mathbf{A})^2 + \mathrm{variance}(\mathbf{A}).
\end{aligned} \tag{SM39}
$$

We begin with the variance term. Using the SVD of $\mathbf{A}$ on the variance term gives,

$$\mathrm{variance}(\mathbf{A}) = \tfrac{\sigma^2}{n}\mathrm{tr}\left(\boldsymbol{\Sigma}^4\bar{\boldsymbol{\Sigma}}^{-4}\right). \tag{SM40}$$

As a consequence of Eqn. SM4 and Eqn. 7,

$$\boldsymbol{\Sigma}_{\mathbf{C}}^2 \preceq \boldsymbol{\Sigma}^2. \tag{SM41}$$

As a consequence of Eqn. SM4 and Eqn. 10,

$$\bar{\boldsymbol{\Sigma}}^{-2} \preceq \bar{\boldsymbol{\Sigma}}_{\mathbf{C}}^{-2} \preceq \frac{1}{1 - (\alpha + 1)\epsilon}\bar{\boldsymbol{\Sigma}}^{-2}, \tag{SM42}$$

where $\bar{\boldsymbol{\Sigma}}_{\mathbf{C}}^2 = \boldsymbol{\Sigma}_{\mathbf{C}}^2 + \tfrac{1}{k}\|\mathbf{C}_{\setminus k}\|_F^2\mathbf{I}$. Because the matrices in Eqn. SM41 and Eqn. SM42 are diagonal and (semi-) positive definite, the relationships can be squared. Finally, because the product of $\frac{1}{(1-(\alpha+1)\epsilon)^2}\boldsymbol{\Sigma}^4\bar{\boldsymbol{\Sigma}}^{-4}$ is bigger or equal to the product of $\boldsymbol{\Sigma}_{\mathbf{C}}^4\bar{\boldsymbol{\Sigma}}_{\mathbf{C}}^{-4}$ for every element along the diagonal,

$$\mathrm{variance}(\mathbf{C}) \leq \frac{1}{(1 - (\alpha + 1)\epsilon)^2}\mathrm{variance}(\mathbf{A}). \tag{SM43}$$

Next we analyze the bias. This part of the proof follows the structure of the proof of Theorem 1 in Alaoui and Mahoney (2015). It will be useful to analyze the quantity,

$$
\begin{aligned}
\|(1-\epsilon)\mathbf{K}(\mathbf{C})\mathbf{y}^*\|_2 &= \|\left((1-\epsilon)\mathbf{K}(\mathbf{C}) - \mathbf{K}(\mathbf{A}) + \mathbf{K}(\mathbf{A})\right)\mathbf{y}^*\|_2 \\
&\leq \|\left((1-\epsilon)\mathbf{K}(\mathbf{C}) - \mathbf{K}(\mathbf{A})\right)\mathbf{y}^*\|_2 + \|\mathbf{K}(\mathbf{A})\mathbf{y}^*\|_2,
\end{aligned}
$$
$$\text{(SM44)}$$

where the last line is due to the triangle inequality. Furthermore,

$$(1-\epsilon)\mathbf{K}(\mathbf{C}) - \mathbf{K}(\mathbf{A}) = \mathbf{K}(\mathbf{C})\left((1-\epsilon)\mathbf{K}(\mathbf{A})^{-1} - \mathbf{K}(\mathbf{C})^{-1}\right)\mathbf{K}(\mathbf{A}), \tag{SM45}$$

so

$$
\begin{aligned}
\|\left((1-\epsilon)\mathbf{K}(\mathbf{C}) - \mathbf{K}(\mathbf{A})\right)\mathbf{y}^*\|_2 &= \\
= \|\mathbf{K}(\mathbf{C})\left((1-\epsilon)\mathbf{K}(\mathbf{A})^{-1} - \mathbf{K}(\mathbf{C})^{-1}\right)\mathbf{K}(\mathbf{A})\mathbf{y}^*\|_2 & \\
\leq \|\mathbf{K}(\mathbf{C})\left((1-\epsilon)\mathbf{K}(\mathbf{A})^{-1} - \mathbf{K}(\mathbf{C})^{-1}\right)\|_{\mathrm{op}}\|\mathbf{K}(\mathbf{A})\mathbf{y}^*\|_2, &
\end{aligned}
$$
$$\text{(SM46)}$$

where op refers to the operator norm $\|\mathbf{A}\|_{op} = \inf\{c \geq 0 : \|\mathbf{A}\mathbf{v}\| \leq c\|\mathbf{v}\| \forall \mathbf{v} \in V\}$, where $V, W$ are a normed vector spaces and $\mathbf{A}$ is a linear map from $\mathbf{A} : V \to W$. We also have,

$$\|\mathbf{K}(\mathbf{C})\left((1-\epsilon)\mathbf{K}(\mathbf{A})^{-1} - \mathbf{K}(\mathbf{C})^{-1}\right)\|_{\mathrm{op}}^2 =$$

$$= ||\mathbf{K}(\mathbf{C}) \left( (1 - \epsilon)\mathbf{K}(\mathbf{A})^{-1} - \mathbf{K}(\mathbf{C})^{-1} \right)^2 \mathbf{K}(\mathbf{C})||_{\text{op}}. \quad \text{(SM47)}$$

From Eqn. SM37, we have

$$(1 - \epsilon)\mathbf{K}(\mathbf{A})^{-1} - \mathbf{K}(\mathbf{C})^{-1} \quad \preceq \quad \frac{\alpha\epsilon}{k}||\mathbf{A}_{\backslash k}||_F^2\mathbf{I}, \quad \text{(SM48)}$$

which is squareable because $(1 - \epsilon)\mathbf{K}(\mathbf{A})^{-1} - \mathbf{K}(\mathbf{C})^{-1}$ commutes with the identity. So,

$$||\mathbf{K}(\mathbf{C}) \left( (1 - \epsilon)\mathbf{K}(\mathbf{A})^{-1} - \mathbf{K}(\mathbf{C})^{-1} \right) ||_{\text{op}}^2$$
$$\leq \frac{\alpha^2\epsilon^2}{k^2}||\mathbf{A}_{\backslash k}||_F^4||\mathbf{K}(\mathbf{C})^2||_{\text{op}}$$
$$\leq \frac{\alpha^2\epsilon^2}{k^2}||\mathbf{A}_{\backslash k}||_F^4||\mathbf{K}(\mathbf{C})||_{\text{op}}^2$$
$$\leq \frac{\alpha^2\epsilon^2||\mathbf{A}_{\backslash k}||_F^4}{||\mathbf{C}_{\backslash k}||_F^4}.$$
$$\leq \frac{\alpha^2\epsilon^2}{(1 - \alpha\epsilon)^2}. \quad \text{(SM49)}$$

The second to last line follows from the definition of the operator norm and $\mathbf{K}(\mathbf{C})$. The last line follows from Eqn. SM36.

Returning to Eqn. SM44 gives,

$$||\mathbf{K}(\mathbf{C})\mathbf{y}^*||_2 \quad \leq \quad \frac{1}{(1 - \epsilon)} \left( \frac{\alpha\epsilon}{(1 - \alpha\epsilon)} + 1 \right) ||\mathbf{K}(\mathbf{A})\mathbf{y}^*||_2$$
$$||\mathbf{K}(\mathbf{C})\mathbf{y}^*||_2 \quad \leq \quad \frac{1}{(1 - \epsilon)(1 - \alpha\epsilon)}||\mathbf{K}(\mathbf{A})\mathbf{y}^*||_2. \quad \text{(SM50)}$$

Therefore, using Eqn. SM36 again,

$$\text{bias}(\mathbf{C}) \leq \frac{1}{(1 - \epsilon)^2(1 - \alpha\epsilon)^2}\text{bias}(\mathbf{A}). \quad \text{(SM51)}$$

Combining Eqn. SM43 and Eqn. SM51 gives,

$$\mathcal{R}(\hat{\mathbf{y}}_{\mathbf{C}}) \leq \max \left( \left( \frac{1}{1 - (\alpha + 1)\epsilon} \right)^2, \left( \frac{1}{1 - \alpha\epsilon} \right)^2 \left( \frac{1}{1 - \epsilon} \right)^2 \right) \mathcal{R}(\hat{\mathbf{y}}_{\mathbf{A}}). \quad \text{(SM52)}$$

On the interval $0 < \epsilon < \frac{1}{2\alpha}$, and for $\alpha > 1$,

$$\max \left( \left( \frac{1}{1 - (\alpha + 1)\epsilon} \right)^2, \left( \frac{1}{1 - \alpha\epsilon} \right)^2 \left( \frac{1}{1 - \epsilon} \right)^2 \right)$$
$$< 1 + \frac{2\alpha(-1 + 2\alpha + 3\alpha^2)}{(1 - \alpha)^2}\epsilon. \quad \text{(SM53)}$$

Theorem 4 follows immediately.

## 1.8 Proof of Theorem 5

The proof is nearly identical to the proof of Theorem 3 of Papailiopoulos et al. (2014), so we do not repeat all of the algebra. It will be convenient to change notation. Without loss of generality, we can sort our column indices $i$ in order of decreasing leverage score. We will also start this index at 1 instead of 0. With this change of notation, the power-law decay formula (Eqn. 15) becomes,

$$\bar{\tau}_i(\mathbf{A}) = (i)^{-a}\bar{\tau}_1(\mathbf{A}) \qquad a > 1, \quad \text{(SM54)}$$

By the definition of the sum of ridge leverage scores and power-law decay (Eqn. SM54),

$$\bar{t} = \bar{\tau}_1(\mathbf{A}) \sum_{i=1}^{d} i^{-a} \quad \rightarrow \quad \bar{\tau}_1 = \frac{\bar{t}}{\sum_{i=1}^{d} i^{-a}}. \quad \text{(SM55)}$$

By the definition of the DRLS algorithm, we collect $|\Theta|$ leverage scores such that $\bar{t} - \epsilon < \sum_{i \in \Theta} \bar{\tau}_i(\mathbf{A})$. This gives,

$$\bar{t} - \epsilon < \bar{\tau}_1 \sum_{i=1}^{|\Theta|} i^{-a} = \frac{\bar{t}}{\sum_{i=1}^{d} i^{-a}} \sum_{i=1}^{|\Theta|} i^{-a} \ \rightarrow \ \epsilon = \bar{t} \frac{\sum_{i=|\Theta|+1}^{d} i^{-a}}{\sum_{i=1}^{d} i^{-a}}. \tag{SM56}$$

Papailiopoulos et al. (2014) show that,

$$\frac{\sum_{i=|\Theta|+1}^{d} i^{-a}}{\sum_{i=1}^{d} i^{-a}} \leq \max\left(\frac{2}{(|\Theta+1)^a}, \frac{2}{(a-1)(|\Theta|+1)^{a-1}}\right) \qquad a > 1. \tag{SM57}$$

Noting that $\bar{t} \leq 2k$, substituting Eqn. SM57 into Eqn. SM56, solving for $|\Theta|$, and noting that the algorithm collects a minimum of $k$ columns results in the expression for the number of columns collected when ridge leverage scores exhibit a power-law decay (Eqn. 15).

## Figures

Figure 1: Pie chart showing multi-omic feature types of matrix $\mathbf{A}$ for LGG tumor data.

Figure 2: Eigenvalues of matrix $\mathbf{A}\mathbf{A}^T$ for LGG tumor multi-omic data. The eigenvalues range over multiple orders of magnitude.

Figure 3: SVD projection of LGG tumor multi-omic data, colored by the combined status for "IDH" and "codel" outcome variables.

Figure 4: SVD projection of LGG tumor multi-omic data, colored by the combined status for "IDH" and "codel" outcome variables.

Figure 5: SVD projection of LGG tumor multi-omic data, colored by the combined status for "IDH" and "codel" outcome variables.

Figure 6: The error $\tilde{\epsilon}$ vs. the number of columns kept for LGG tumor multi-omic data $k = 3$ ridge leverage scores. A dramatic reduction in the number of columns kept incurs only a small error penalty.

Figure 7: Power-law decay of LGG tumor multi-omic data $k = 3$ ridge leverage scores with sorted column index. The fit is to Score = b $\times$ (Index) $^a$ on the first $10^3$ ridge leverage scores.

Figure 8: Histogram of $||\mathbf{C} - \mathbf{XC}||_F^2/||\mathbf{A} - \mathbf{XA}||_F^2$ for 1000 random rank-$k = 3$ orthogonal projections $\mathbf{X}$ and $\mathbf{C}$ selected by the $k = 3, \epsilon = 0.1$ DRLS algorithm for LGG tumor multi-omic data.

Figure 9: Classical leverage scores vs. $k = 3$ ridge leverage scores for LGG tumor multi-omic data. Most columns with large classical leverage scores have smalll ridge leverage scores; there is significant shrinkage.

Figure 10: Histogram of $\hat{\mathbf{y}}_{\mathbf{A}} - \hat{\mathbf{y}}_{\mathbf{C}}$ for the outcome "codel."

Figure 11: Histogram of $\hat{\mathbf{y}}_{\mathbf{A}} - \hat{\mathbf{y}}_{\mathbf{C}}$ for the outcome "IDH."

Figure 12: Histogram of $\hat{\mathbf{x}}_{\mathbf{A}} - \hat{\mathbf{x}}_{\mathbf{C}}$ for the outcome "codel."

Figure 13: Histogram of $\hat{\mathbf{x}}_{\mathbf{A}} - \hat{\mathbf{x}}_{\mathbf{C}}$ for the outcome "IDH."

Figure 14: Histograms of the predictions $\hat{\mathbf{y}}_{\mathbf{C}}$ conditioned the outcome $\mathbf{y}$ for "codel."

Figure 15: Histograms of the predictions $\hat{\mathbf{y}}_\mathbf{C}$ conditioned on the outcome $\mathbf{y}$ for "IDH."