[Reviews · NeurIPS 2018]

Reviewer 1



Ridge regression is an alternative for linear regression on low-rank matrix inputs, and provides a regularization for ill-posed problems. Ridge leverage scores and rank-k subspace leverage scores (display equations 1 and 2 in the paper) have been at the core of many algorithms for matrix sketching and row/column subset selection. The authors extend an deterministic algorithm by Papailiopoulos et al. (2014) on rank-k subspace scores sampling and make it work for ridge leverage scores. The motivation is that rank-k subspace scores ignore the small principal components, whereas regularization given by ridge leverage scores is a better, natural, stable alternative. The authors use this to get deterministic algorithms for column subset selection, projection-cost preservation etc. They show this under an assumption of power-law decay of ridge leverage scores, along the lines of a similar assumption used in Papailiopoulos et al. Overall, most of the key ideas have already been there in previous work by Cohen et al., Papailiopoulos et al. etc. I find this work incremental and borderline for NIPS.

Reviewer 2



--------------------------After rebuttal---------------------------- I maintain my score. I also strongly approve of the suggestion of highlighting more the experimental results in the main paper. ----------------------------------------------------------------------- The paper describes a novel deterministic selection rule based on ridge leverage scores (RLS). RLSs are well known quantities used in randomized sketching and coreset selection to identify influential samples. Similarly, it is known that sampling **and reweighting** rows of a matrix A according to their RLS produces a sketch that whp approximates the true matrix up to a small multiplicative and additive error. The authors prove that sorting the rows in descending order (by RLS), and deterministically selecting them until the RLS falls under a carefully chosen threshold is also sufficient to obtain a provably accurate sketch. Therefore, they propose deterministic RLS selection as a convenient and provably accurate rule for column selection, with improved interpretability over optimization based alternative such as lasso and elastic net. Contribution: The additive-multiplicative spectral bound (Theorem 1) this is the core contribution of the paper. Note that Thm. 2,3,4 are direct consequences of Thm. 1, and are obtained with a straightforward application of results from (Boutsidis et al. 2011), (Cohen et al. 2015), (Cohen et al. 2017), (Papailiopoulos et al. 2014) and (Alaoui and Mahoney, 2015). Similarly, Thm. 5 is almost a direct consequence of (Papailiopoulos et al. 2014). This, together with the only two modifications necessary to existing results, is explained in detail in the complete appendix, which matches and exceed NIPS's standards for clarity. Note that it is probably possible to leverage this method into even more results, for example statistical risk guarantees stronger than Thm 4 using results from Rudi, Alessandro, Raffaello Camoriano, and Lorenzo Rosasco. "Less is more: Nyström computational regularization." Advances in Neural Information Processing Systems. 2015. While the paper introduces a new method with several small improvement over (Papailiopoulos et al. 2014), the work remains overall incremental and the significance of the contribution from a technical point of view is limited. The authors should try to stress more and clarify any significant improvement over (Papailiopoulos et al. 2014), and any advantage of RLS over k-rank LS. For example, RLS methods are widely successful in Reproducing Kernel Hilbert Spaces, while k-rank LS are not. Nonetheless, I still believe this paper could find its audience at NIPS, and weakly suggest acceptance.

Reviewer 3



This paper studies deterministic column sampling using ridge leverage scores. These scores represent a global, operator motivated way of defining importance w.r.t rank-k approximations to the matrix, and have a variety of highly useful theoretical properties. The paper combines these recent works with a result on deterministic column subset selection. It gives significantly better and more general theoretical bounds, using a combination of techniques from ridge leverage scores and matrix algorithms. Then, it demonstrates the good performances of this algorithm on matrix data arising from computational biology. Just as importantly, it demonstrates that the matrices addressed do have power-law decay in eigenvalues, a key assumption in the deterministic column subset selection literature. I find both the theoretical and experimental studies in this paper impressive. It is an excellent demonstration of the power of newer definitions of leverage scores, and provides empirical justification for many underlying assumptions in the bounds. As a result, I strongly advocate for its acceptance.